# An Adaptive Simultaneous Multi-Protocol Extension of CRAFT

**DOI:** 10.3390/s23084074

**Published:** 2023-04-18

**Authors:** Louis Moreau, Emmanuel Conchon, Damien Sauveron

**Affiliations:** 1XLIM (UMR CNRS 7252/Université de Limoges), MathIS, 87060 Limoges, France; 2SAS ICOHUP, 87000 Limoges, France

**Keywords:** computer security, Internet of Things, remote attestation, multi-protocol, smart-city

## Abstract

An exponential number of devices connect to Internet of Things (IoT) networks every year, increasing the available targets for attackers. Protecting such networks and devices against cyberattacks is still a major concern. A proposed solution to increase trust in IoT devices and networks is remote attestation. Remote attestation establishes two categories of devices, verifiers and provers. Provers must send an attestation to verifiers when requested or at regular intervals to maintain trust by proving their integrity. Remote attestation solutions exist within three categories: software, hardware and hybrid attestation. However, these solutions usually have limited use-cases. For instance, hardware mechanisms should be used but cannot be used alone, and software protocols are usually efficient in particular contexts, such as small networks or mobile networks. More recently, frameworks such as CRAFT have been proposed. Such frameworks enable the use of any attestation protocol within any network. However, as these frameworks are still recent, there is still considerable room for improvement. In this paper, we improve CRAFT’s flexibility and security by proposing ASMP (adaptative simultaneous multi-protocol) features. These features fully enable the use of multiple remote attestation protocols for any devices. They also enable devices to seamlessly switch protocols at any time depending on factors such as the environment, context, and neighboring devices. A comprehensive evaluation of these features in a real-world scenario and use-cases demonstrates that they improve CRAFT’s flexibility and security with minimal impact on performance.

## 1. Introduction

More and more everyday objects are now connected to IoT networks and the Internet [1]. Such connected devices can take part in different networks, in particular in smart cities [2,3], such as the one depicted in Figure 1, where connected vehicles, smart lighting, connected traffic lights and other devices can interact with each other. As these devices are everywhere and sometimes even physically accessible to everyone, they make easy targets for attackers. At the same time, these devices are often limited in resources and, thus, limited in security capabilities, usually to reduce their cost or increase their battery life.

In order to secure such a network, a proposed solution is remote attestation. Remote attestation establishes two categories of devices, verifiers and provers. Provers must send an attestation to one or many verifiers when requested or at regular intervals to maintain trust by proving their integrity. Using remote attestation, the device’s firmware is regularly checked against a known good state, and, thus, compromised devices are excluded and the network stays healthy. Existing remote attestation protocols usually focus on specific mechanisms: software protocols that can be applied to any device, hardware protocols with strong hardware requirements, and hybrid protocols with minimal hardware requirements, which rely on some specific software.

To further improve remote attestation flexibility and security, there are also remote attestation frameworks, such as CRAFT [4], which improve existing protocols by adapting to any deployment context, and, thus, are more flexible and secure than standalone remote attestation protocols.

The aim of this paper is to present the ASMP (adaptive simultaneous multi-protocol) features of CRAFT, which further improve flexibility and security as a device can not only use any attestation protocol, but can also switch from one protocol to another at any time, using the most appropriate protocol. To highlight how the CRAFT and the ASMP features enable continuous remote attestation in complex environments containing highly diverse devices better than any existing protocols, this paper simulates a real-world environment represented by a smart city, and uses different scenarios in which devices operate.

### 1.1. Contribution

The salient contributions of this paper are:The simultaneous use of multiple remote attestation protocols in the IoT network, which enables different kinds of devices to interact together.The capability for devices to seamlessly switch, in real time, from one remote attestation protocol to another depending on factors such as the environment, context, or neighbouring devices.The definition of a real-world scenario and use-cases using a smart city representation.An extensive evaluation of CRAFT with and without the ASMP features, as well as a comparison with the combination of standalone attestation protocols SEDA and US-AID. This comprehensive evaluation demonstrates that the ASMP features improve CRAFT flexibility and security with minimal impact on performance.

### 1.2. Structure

Section 2 provides an overview of related work for remote attestation protocols. These remote attestation protocols can be used in a framework with the ASMP features. Section 3 presents the CRAFT framework, which serves as a root for the ASMP features, and details how these features work, as well as their implementation. Finally, Section 4 presents the evaluation methodology and demonstrates the security level and performance level of the ASMP features based on simulation results.

## 2. Related Work

Remote attestation solutions are developed in order to increase IoT network security. Such solutions aim at ensuring that devices are in a known good state (with regard to firmware and memory), so that they can be trusted to take part in network operations.

Most existing remote attestation solutions focus on specific sub-sections of attestation, which fall into three main categories: software attestation, hardware attestation and software-hardware, also called hybrid attestation.

Software attestation [5,6] is defined by the absence of additional hardware requirements. It is usually used for very constrained devices that are limited in performance. Such solutions usually rely on challenge–response mechanisms [7,8].

In contrast, hardware attestation [9,10,11,12] requires additional hardware security features. For example, some of the most commonly used features are TPMs (trusted platform modules) [13], TEEs (trusted execution environments) [14] and PUFs (physical unclonable functions) [15].

The third solution is hybrid attestation, which only requires minimal hardware security features, such as a ROM (read-only memory) and an MPU (memory protection unit) to ensure the correct execution of critical code. Some work on hybrid attestation has focused on the hardware security mechanisms, and some has focused on protocols using this hardware. Hybrid attestation protocols are the most relevant to compare to remote attestation frameworks as presented in this paper. Thus, the remainder of this section focuses on related hybrid attestation solutions.

The first published solution to perform hybrid attestation was SMART (2012) [16]. SMART uses low-cost hardware (i.e., the combination of a ROM and an MPU) to store critical code and keys and to ensure that keys are only read by authorized code, and that critical code can only be executed for its intended purpose.

Trustlite (2014) [17] extends SMART, with larger access control rules for the MPU. It takes into account different kinds of memory hardware, such as DRAM and Flash, as well as peripherals, such as timers. Trustlite also introduces the use of OS and trusted states called Trustlets.

SEDA (2015) [18] was one of the first solutions to define a hybrid-based swarm attestation protocol. SEDA implementation makes use of SMART and Trustlite. In SEDA, attestation is requested by a trusted verifier from a device, which transmits it to its neighbours and so on, creating a spanning tree. In the end, the initial device receives the sum of all responses and transmits it to the trusted verifier. Responses can contain different levels of detail, ranging from the number of compromised devices to the full attestation list of all devices. In later proposals, such as DARPA [19] and US-AID [20], SEDA has been criticized by its own authors for not being adequate for real-world applications as it only focuses on remote software adversaries and a specific network model (swarms). However, it is used as a basis for comparison in many other studies because of its anteriority and broad application scope.

DARPA (2016) [19] uses a heartbeat mechanism, which sends periodic messages to neighbouring devices in order to ensure that they are not taken out of the network. This mechanism enables the detection of physical attacks as it assumes that, to physically attack a device, the attacker must disconnect it from the network for a significant amount of time (leaving aside side-channel attacks).

US-AID (2018) [20] uses PONAs (proofs of non-absence), which are similar to heartbeats. PONAs are used to show that devices are continuously connected to the network, and have not been compromised through a physical attack. Using PONAs, devices can move, under the restriction that their future neighbour is next to their previous neighbour.

SIMPLE (2020) [21] is not exactly a hybrid attestation solution, but still has stronger requirements than the usual software attestation solutions. It does not require additional hardware security features and, thus, encompasses more IoT devices. Its security comes from Security MicroVisor, a formally verified software-based memory isolation technique. It separates untrusted application software from a so-called trusted computing module.

CRAFT (2021) [4] is the first continuous remote attestation framework for the IoT. It provides all the basic functionalities needed to implement continuous remote attestation in real-world IoT networks. CRAFT is agnostic to the attestation protocols used, and its flexibility enables it to be used in both static and mobile networks. Moreover, CRAFT is open to upgrades and extensions.

Helble et al. (2021) [22] promote the need for flexible attestation mechanisms, which are closer to the CRAFT framework paradigm than regular remote attestation protocols. They propose features such as protocol flexibility, which is implicitly included in the ASMP features, and policy-based negotiation for attestations protocols, which is complementary to the ASMP features. To do so, they use Copland from Ramsdell et al. [23], a dedicated specification language, to provide several attestation protocols examples, which may represent a future improvement for our framework.

CFRV (2022) [24] improves the existing control-flow attestation schemes using a slice mechanism. It enables a secure decentralized CFA mechanism where both prover and verifier check each other, without the need of a single trusted device. Moreover, CFRV is able to reduce the time consumed by cryptographic operations as the public key infrastructure is only used during the registration of a device.

SCRAPS (2022) [25] uses blockchain as a proxy to enable many-to-many attestation between verifiers and provers. The attestation mechanism follows a publish-subscribe logic: provers publish their attestation in the blockchain, while verifiers act as subscribers and receive attestations. Thus, SCRAPS enables multiple verifiers, and its asynchronicity allows on-demand attestation retrieval, while minimizing the device’s electrical consumption.

## 3. Craft Contextual Adaptation to Multiple Protocols

The following section of the paper describes how the ASMP (adaptive simultaneous multi-protocol) features work in CRAFT, and how this demonstrates CRAFT flexibility while improving security in comparison to other standalone protocols. The new features of this work are described in Section 3.1. Section 3.2 details how CRAFT and the attestation protocols interact together, and adds details needed to provide these features. This new implementation is then evaluated in Section 4 with regard to both performance and security.

### 3.1. An Overview of CRAFT New Features: CRAFT-ASMP

The CRAFT [4] framework enables continuous remote attestation for IoT networks and agnostically supports any attestation protocol.

This work aims to demonstrate how CRAFT can implement the ASMP features to integrate any remote attestation protocol into the framework. These features enable the framework to seamlessly switch between multiple remote attestation protocols, and, thus, use the most appropriate protocol depending on the current context.

Context definition includes elements such as the network a device is in, what other devices a given device is communicating with at the moment, the physical environment (indoor or outdoor), and current parameters, such as the speed and temperature. Being able to select and seamlessly switch protocols depending on the context enables maximization of security while minimizing performance overheads at any time during the network’s life.

In the following section, CRAFT using ASMP features is called CRAFT-ASMP. This work also illustrates how CRAFT-ASMP behaves in a real-world scenario where multiple attestation protocols are used at the same time. In this scenario, some devices use real-time attestation protocol adaptation and can switch from one attestation protocol to another thanks to these novel CRAFT capabilities. Finally, CRAFT-ASMP enables better security performance with regard to the exclusion of compromised devices compared to standalone attestation protocols. Moreover, CRAFT-ASMP adds little to no overhead compared to CRAFT, as demonstrated by the simulation results in Section 4.

### 3.2. Switching between Multiple Remote Attestation Protocols

Before detailing the ASMP features and the remote attestation protocols chosen as a point of comparison in the evaluation, the original CRAFT framework is overviewed.

#### 3.2.1. CRAFT Basis

CRAFT is a continuous remote attestation framework for IoT networks, which uses several parameters to achieve its purpose. A general overview of CRAFT is provided in Figure 2. In the following paragraphs, the main ideas of the CRAFT [4] framework are presented to aid understanding.

In a nutshell, CRAFT has two phases: the offline phase and the online phase. Configuration and initialization happen during the offline phase, while communication within the network happens during the online phase. During the online phase, devices are split between *K*- and *L*-devices (i.e., respectively, the core network and the outer network). *L*-devices communicate with *K*-devices using several messages detailed below, the most commonly used being heartbeats. Heartbeats are what increase the communication frequency and, thus, trust between devices, increasing security at a lower performance cost than a full attestation.

The minimal parameters used by CRAFT to define a device $D_{i}$ are its identity $id_{i}$, its security strength $s_{i}$, its maximum mobility $h_{i}$, and attestation and heartbeat timers $T_{a_{i}}$ and $T_{b_{i}}$. Thus, a device $D_{i}$ is defined with the parameters set $P_{i}=\left\{id_{i},s_{i},h_{i},T_{a_{i}},T_{b_{i}}\right\}$. These parameters are provided by an operator *O* during the offline phase. The security strength $s_{i}$ of the device is established by *O* depending on the deployment context and the device hardware security features. According to the value of $s_{i}$, devices belong either to the core network and are *K*-devices, or to the outer network and are *L*-devices. If $s_{i}$ is too low, devices cannot participate in the network. More details about network configuration in CRAFT can be found in [4].

Once a device is configured it can join the network and the online phase starts. During this phase, devices exchange messages named connect, beat, attest, lost and reconnect. These messages share a common header as depicted in Figure 3.

The connect message lets a device start communication with new neighbours, provided the device is properly authorized. Heartbeat messages maintain frequent communication between *K*-devices and *L*-devices, improving security by ensuring the continuous attestation of the network, in addition to the attest messages.

The attest messages’ format and features depend on the underlying attestation protocols available to the devices, as the CRAFT framework only encapsulates them.

lost is the message sent by devices when they cannot communicate with a neighbour anymore. By sending it, they signal to the network that their neighbour might have moved, and that the network should expect a reconnect message. If the device tries to reconnect too long after moving (i.e., when the disconnection time exceeds a threshold), the network will not accept it back since the moving device is considered as having been compromised.

As CRAFT is a flexible framework, it can easily add features, such as the ASMP features described next.

#### 3.2.2. CRAFT ASMP Features in Depth

In order to enable the selection of the attestation protocol according to the device status, the parameters field of each message is used. This field is defined in Figure 4. The current implementation uses a 64 bit bitfield, numbered from 0 to 63. The leftmost bits labelled (1) are set to 1 when the corresponding protocol is available to the device (e.g., when bits 0 and 1 are set, both attestation protocols P1 and P2 can be used, if it is considered that the framework only supports two protocols). The same number of bits that are in (1) are used to set the preferred protocol in (3). When a bit is set to 1 in (3), the corresponding available protocol in (1) is currently preferred (e.g., when bit 32 is set, the device will preferentially use attestation protocol P1). The preferred protocol can change, which enables devices to switch to the most appropriate attestation protocol at any time, depending on the device context. Bit 31 labelled (2) is set to 1 when the given device is a *K*-device. Finally, bits labelled (r) are reserved for future use (i.e., for future additional attestation protocols).

#### 3.2.3. Featured Remote Attestation Protocols

The capabilities of CRAFT using the ASMP features are demonstrated using the two existing remote attestation protocols, US-AID [20] and SEDA [18]. Since they have distinct characteristics, these two protocols are interesting to use alongside CRAFT, and as standalone points of comparison.

##### US-AID [20]

US-AID is an interesting attestation protocol to include in this evaluation. A standalone, US-AID provides heartbeats, as does CRAFT, which enables better comparison. US-AID’s attestation mechanism has the specificity to not be distributed to the whole network, and works particularly well with mobile networks. In the evaluation, only the attestation features of US-AID are kept.

##### SEDA [18]

The SEDA attestation protocol provides different relevant features: it provides a fairly standard remote attestation protocol that spans the attestation result of the entire network back to the device that initiated the attestation process. However, while it provides a good point of comparison for static environments, SEDA is not efficient with mobile devices, and lacks CRAFT’s flexibility and performance.

These two protocols enable CRAFT-ASMP features to be evaluated, as devices integrate both SEDA and US-AID attestation protocols and can switch between them according to the context. The US-AID heartbeat feature can also be compared to CRAFT’s with regard to performance.

## 4. Evaluation

In order to demonstrate that CRAFT-ASMP improves security and has good performance, it was evaluated in the Omnet++ simulation framework [26]. In these simulations, CRAFT-ASMP was compared to regular CRAFT and to the so-called SEDA+AID framework, which includes both SEDA and US-AID raw attestation protocols, as presented in Section 4.1. This shows how these new features improve CRAFT, and how CRAFT-ASMP outperforms the SEDA+AID framework defined below.

Section 4.1 further details the frameworks, environment and scenarios used in the simulation, including their parameters and which types of devices are used. Section 4.2 describes the metrics analyzed as well as the methodology used to obtain significant results with good confidence. Finally, Section 4.3 presents simulation data and provides an analysis, demonstrating CRAFT-ASMP’s usefulness in terms of security and efficiency.

### 4.1. Frameworks and Scenarios Description

#### 4.1.1. Frameworks

Three different frameworks are defined for the simulations:CRAFT-ASMP: this is the new CRAFT framework with the ASMP (adaptive simultaneous multi-protocol) features. Some devices can switch between supported attestation protocols depending on the context. All CRAFT features are available to all devices.CRAFT: this is the original CRAFT framework without the ASMP features. Contrary to CRAFT-ASMP, all devices only support a single attestation protocol available through the simulation, even though several protocols are supported by CRAFT.SEDA+AID: SEDA and AID attestations coexist within the same network but there is no direct connection between them. Devices from the core network can handle both protocols but devices with different protocols cannot interact. As such SEDA+AID is considered as a framework for evaluation purposes.

#### 4.1.2. Environment

The environment in which the scenarios take place is a smart city represented as a grid, as depicted in Figure 5. Here, only a representative part of the simulation environment is represented. The real-world view represents physical devices, such as buildings, vehicles and smart lighting. The same devices, and their communication with each other, are represented in the logic view. Using such a smart city representation enables clear highlighting of the benefits of CRAFT and in particular the ASMP features. Indeed, devices are very heterogeneous as they would be in real life, both in their mobility and security performances. The full characteristics of the simulation environment are described in Table 1. In particular, the smart city is represented as a square of 6 by 6 smaller square blocks. Each block is 100 m by 100 m, which gives the whole simulation dimensions of 600 m by 600 m.

##### Devices

In all scenarios, devices placed in the smart city are of three types:Antennas, which do not move on the grid and are placed so they cover the whole grid. In CRAFT-ASMP and CRAFT, they are *K*-devices and form the core network.Static devices, randomly placed on the grid. They illustrate connected devices, such as smart lighting and air pollution sensors. In CRAFT-ASMP and CRAFT, they are *L*-devices and part of the outer network.Mobile devices, which follow the lines of the grid. They are placed on random intersections at the beginning of the simulation, move along the lines of the grid, and select a random direction (forward, backward, left or right) when they arrive at the next intersection. These devices mimic vehicles. In CRAFT-ASMP and CRAFT, they are *L*-devices and part of the outer network.

##### Scenarios

To test different aspects of the frameworks, three scenarios are used; their Omnet++ simulation parameters are listed in Table 1. Specific scenario parameters are listed in Table 2. The three scenarios are:The *Basic* scenario is the smart city only, in which all devices demonstrate how they behave when there is no particular event. It will be used as a point of comparison with other scenarios. The default parameters, in particular, timings and distances, used in this simulation are arbitrarily chosen, such that they are representative of a real-world scenario.The *Parking* scenario is similar to the *Basic* scenario except that a “parking” phase is added. During this phase, mobile devices stop moving on the grid for a certain time to simulate vehicles that are parked. For practical simulation purposes, all devices stop for the same duration after a common delay.The *Parking+Out* scenario adds an “out time” to the *Parking* scenario: some mobile and static devices stop communicating in the network and will try to reconnect later. Half of these devices have a short “out time” of 2000 s that simulates genuine disconnection and are expected to successfully rejoin the network. The other half have a long “out time” of 7200 s that simulates an attack; these devices are not supposed to be accepted back into the network by the antennas (i.e., the core network) as this duration exceeds a $T_{a_{i}}$ duration of 3600 s.

To summarize, in this smart city environment, these three frameworks will be run on the three scenarios, resulting in nine different simulations, and, thus, nine sets of results to compare.

### 4.2. Metrics and Methodology

To demonstrate the efficiency of CRAFT-ASMP, five metrics are observed:Devices exclusion: a good framework must be sensitive and exclude all compromised devices by maximizing the true positive rate. Exclusion must also be specific and non-compromised devices must be maintained in the network by minimizing the false positive rate.Attestations count: attestations depend on the attestation protocol and the attestation count should correlate with the frequency defined in the parameters. With SEDA, the attestation count should always be the same (e.g., 23 in a 24 h simulation with attestations every hour). With US-AID, the attestation count depends on the number of neighbour devices, and differences can be explained by the framework definition of a neighbour device or by different contexts. Each device should have at least one attestation per attestation period, regardless of the attestation protocol.Heartbeats count: heartbeats help to improve continuous attestation with an increased frequency but lower performance impact than attestation messages. The more heartbeats there are per device, the more secure the network is.Average data volume: all frameworks should minimize their impact on the device, and the data volume exchanged by the framework is one point of comparison. Sending and receiving data is a source of power consumption and frameworks should minimize such overheads.HMAC computations: like data volume, cryptography has an impact on device performances and power consumption. It should be minimized, but not at the expense of security.

The methodology used in this work to provide significant measurements, as well as confidence intervals around average values, is the following:

Each simulation of a framework in a scenario is repeated 100 times using a different seed in each repetition to be used by the Omnet++ random number generator. For example, the *Basic* scenario using CRAFT-ASMP is repeated 100 times using 100 different seeds, and the *Parking* scenario using SEDA+AID is also repeated 100 times using the same seeds. Using the simulation results, the assumption is made that all measurements follow a normal distribution.

To validate this assumption for a given metric, 100 repetitions of a given scenario are evaluated with two graphical methods: histograms and Q-Q plots. Histograms show the frequency of a metric’s values while Q-Q plots compare a metric values’ quantiles to the theoretical quantiles of a normal distribution. If the assumption holds, histograms are expected to be roughly normal-shaped and Q-Q plots are expected to follow a straight line.

This methodology is applicable to all presented results for any metrics in any of the nine simulations of this work. However for readability and conciseness reasons, in the following, this methodology is only shown for two metrics: the average data volume sent by devices in the *Basic* scenario using the CRAFT-ASMP framework, and the false positive mobile devices exclusions in the *Parking+Out* scenario using the CRAFT-ASMP framework. This is depicted in Figure 6 using histograms and Q-Q plots.

The histogram in Figure 6a is very close to a normal distribution, and is only slightly asymmetrical. Moreover, the Q-Q plot in Figure 6b mostly follows a straight line with minor irregularities. Similarly, the histogram in Figure 6c is close to a normal distribution. However, the fact that this metric is discrete with a minimum of 0 makes it a bit less obvious, especially as it is not symmetrical on the left side. The Q-Q plot in Figure 6d shows the same information with a straight line except for the first point.

The conclusion is that, even though the data do not strictly follow a normal distribution, we can use the central limit theorem. As simulations are repeated 100 times each, results will be well approximated by a normal distribution. Thus, confidence intervals can be calculated using Student’s *t*-distribution and Equation (Equation 1) with a α=95% confidence limit and the number of repetitions n=100 and *t* taken from the Student’s *t* table [27].
(1)$$\left[\;\bar{X}-t_{\alpha /2}^{n-1}\frac{S}{\sqrt{n}};\bar{X}+t_{\alpha /2}^{n-1}\frac{S}{\sqrt{n}}\;\right]$$
$$\begin{array}{c} \mathrm{with}\;\mathrm{the}\;\mathrm{sample}\;\mathrm{mean}\;\bar{X}=\frac{1}{n}\sum_{i=1}^{n}X_{i}\\ \mathrm{and}\;\mathrm{the}\;\mathrm{sample}\;\mathrm{variance}\;S^{2}=\frac{1}{n-1}\sum_{i=1}^{n}{\left(X_{i}-\bar{X}\right)}^{2} \end{array}$$

This enables the results presented in the following section to be easily verified and repeated.

### 4.3. Results

Previously mentioned metrics (i.e., device exclusions, attestation counts, heartbeat counts, average data volume and HMAC computations) are analysed in this section. These results demonstrate that CRAFT-ASMP greatly improves CRAFT and SEDA+AID in several ways. Indeed, CRAFT-ASMP demonstrates its security as it excludes correctly compromised devices better than other frameworks. In addition, CRAFT-ASMP demonstrates better performances than other frameworks with regard to computation and communication.

#### 4.3.1. Devices Exclusion

Monitoring device exclusions is critical to assess the security of an attestation framework as all compromised devices must be excluded while maintaining other devices in the network.

In Table 3, the true positive and false positive rates are shown.

True positives can only happen for excluded devices in the *Parking+Out* scenario as devices do not simulate exclusion in other scenarios. The results indicate that, using CRAFT with or without the ASMP features, 100% of attacked devices are excluded. However, as SEDA+AID uses standalone protocols, SEDA does not perform compromised static devices exclusions (which is depicted by a 0±0% exclusion value in Table 3), so only US-AID mobile devices are excluded.False positives can happen for mobile devices when they move outside the network and are not able to reconnect to it quickly enough (i.e., before the scenario-specific threshold). Results shown both in Table 3 and in Figure 7 demonstrate that both CRAFT implementations behave in a similar manner, whereas SEDA+AID wrongfully excludes far more devices.

These better results for CRAFT and CRAFT-ASMP are explained by the use of the lost packet, which enables devices to rejoin the network more easily than in US-AID.

#### 4.3.2. Attestations and Heartbeats

Attestations and heartbeats define how frequently a device is checked by the network. A device should be checked frequently enough to maintain trust in it. Thus, more attestations and heartbeats are better for security, but these messages must not add too much communication and computation overhead to devices.

Figure 8 shows how many US-AID attestations are exchanged among mobile devices in all scenarios.

In the *Basic* scenario, the SEDA+AID framework has more attestations per device than CRAFT and CRAFT-ASMP because the US-AID protocol usually has more neighbours at a given time than CRAFT. Indeed, it is more permissive from this perspective than CRAFT, and attestations are sent to all neighbours with the US-AID protocol.In the *Parking* and *Parking+Out* scenarios, CRAFT-ASMP mobile devices switch to SEDA when parked, which represents 37.5% of the scenario duration, so they have fewer US-AID attestations. This explains why the amount of attestations for CRAFT-ASMP in the *Parking* scenario is about 46.1% less than the amount of attestations for CRAFT-ASMP in the *Basic* scenario (from 29.05 to 63.05). This does not happen in other frameworks as they do not switch attestation protocols.Across all three scenarios, CRAFT and SEDA+AID stay at similar attestation levels.However, the *Parking* scenario has respectively a +10.3% (from 63.05 to 69.55) and +2.0% (from 79.61 to 81.24) attestations count difference compared to the *Basic* scenario. This difference is explained by the reduced mobility of parked devices.Similarly, the *Parking+Out* has +1.6% (from 63.05 to 64.06) for the CRAFT and −10.8% (from 79.61 to 71.05) for the SEDA+AID attestations count difference. This difference is explained by the combination of parked and excluded devices.

Thus, when comparing CRAFT-ASMP to CRAFT or SEDA+AID regarding attestations, the usefulness of the ASMP features is clear.

Table 4 shows how many SEDA attestations are exchanged among static and mobile devices in all scenarios (when applicable).

For SEDA attestations, all static devices in the *Basic* and *Parking* scenarios showed 23 attestations. As these results are more straightforward, they are not represented in a figure. CRAFT-ASMP in *Parking* and *Parking+Out* scenarios have only 6.39±0.07 and 5.89±0.07 attestations on average as they use US-AID when not parked.In the *Parking+Out* scenario, there are also fewer SEDA attestations per device on average, because of the device exclusions at some point in the network lifetime. That number is higher without CRAFT (21.81±0.00 without versus 21.33±0.00 with) because excluded static devices in SEDA+AID do not stay out of the network and, thus, perform more attestations.

To summarise the attestation count comparisons, all protocols have a comparable number of attestations. However, CRAFT-ASMP demonstrates its superiority in using the most appropriate protocol, depending on the context, to increase the whole network security. Indeed, devices using CRAFT-ASMP can use SEDA attestation when parked, as SEDA is more appropriate than US-AID for static devices and demonstrates better results in this context.

Figure 9 shows the average number of beats performed by each device during the lifetime of the network.

CRAFT-ASMP performs as well as CRAFT, because it provides the same heartbeat mechanism to all devices. Providing heartbeats is an essential security feature that completes full attestations with a lightweight but more frequent device check.Across all three scenarios, the heartbeats count difference between CRAFT-ASMP and CRAFT is less than 0.05%. This demonstrates that CRAFT-ASMP’s additional features have no impact on the performance of the heartbeat mechanism.In comparison to SEDA+AID, CRAFT-ASMP always shows more than a 60% increase in the heartbeats count. This is explained by the fact that only US-AID devices (i.e., 40% of devices) are able to perform PONAs, which are similar to CRAFT’s heartbeats. This clearly shows the benefits of CRAFT-ASMP, as maintaining continuous attestation through heartbeats helps to keep the network secure.

#### 4.3.3. Average Data Volume and HMAC Computations

These two metrics correlate with energy consumption and computation time as they represent most of the frameworks’ workload and computation time. They enable analysis of the general performance of the frameworks.

Figure 10 shows the average amount of data sent by devices.

In all scenarios, CRAFT-ASMP exchanges less data than SEDA+AID, as the messages used are smaller. In the *Basic* scenario, the difference between CRAFT-ASMP and SEDA+AID is 6.5% (from 394,518 to 370,561 bytes).In both the *Parking* and the *Parking+Out* scenarios, the SEDA+AID framework has over a 60% data volume overhead compared to CRAFT-ASMP (respectively, 381,245 compared to 236,929 bytes and 380,323 compared to 231,053 bytes). This is explained by CRAFT-ASMP taking advantage of the reduced mobility, which makes devices exchange far fewer lost packets and, thus, reduces the data volume sent.In the *Parking* scenario, CRAFT-ASMP exchanges 1.2% more data than CRAFT (from 234,083 to 236,929 bytes), which is explained by mobile devices switching to the parked state and using the SEDA attestation. This is a bit less lightweight than US-AID attestation, but this does not show a significant performance impact in the simulations. Moreover, this result demonstrates that, while CRAFT-ASMP can switch to SEDA attestation, which is more efficient for static devices, this does not come at the cost of efficiency.

Table 5 illustrates how data are balanced between antennas, static devices and mobile devices in each framework. The results show that antennas are the main source of communication, always representing over 87% of the data volume.

The balance between antenna and non-antenna devices leans more towards the non-antenna devices when using CRAFT-ASMP and CRAFT than using SEDA+AID (e.g., from 96.04% for SEDA+AID to 87.81% for CRAFT-ASMP and 88.40% for CRAFT in the *Parking* scenario). This is explained by the use of beat and lost packets for all devices, which increase communication from static and mobile devices to the benefit of increased security.In the *Parking* and *Parking+Out* scenarios compared to the *Basic* scenario, there are fewer lost packets and, thus, antennas are less solicited, which explains the reduced weight of antennas in the total data volume sent (e.g., for CRAFT-ASMP, from 92.46% in the *Basic* scenario to 87.81% in the *Parking* scenario).

Figure 11 illustrates the average number of HMACs computed by devices.

CRAFT-ASMP shows similar performance to CRAFT in all scenarios, even with its additional features. CRAFT-ASMP is also more efficient the less mobility there is (and, thus, fewer lost packets): the average number of computed HMACs is reduced by more than 36% (from 11,693 to 7448) between the *Basic* and *Parking* scenarios.The SEDA+AID values show little variation in terms of the average HMAC computations across scenarios. The difference with the *Basic* scenario is less than 4% (from 5990 to 5798 for the *Parking* scenario and to 5761 for the *Parking+Out* scenario), which is due to the reduced mobility. This variation is not greater because the framework does not react to the evolving context.The SEDA+AID framework always uses fewer HMACs than CRAFT-ASMP. There is an overhead of 48.8% (from 5990 to 11,693) for CRAFT-ASMP compared to SEDA+AID in the *Basic* scenario, and an overhead of 22.2% (from 5798 to 7448) for CRAFT-ASMP compared to SEDA+AID in the *Parking* scenario. This overhead is explained by CRAFT-ASMP implementing the additional security of the beat packet in all devices, whereas it is only available to US-AID supporting devices (40% of devices, i.e., the mobile devices) in the SEDA+AID framework.

In summary, the results demonstrate that CRAFT-ASMP is indeed able to improve the security and flexibility of existing remote attestation protocols.

First, it can exclude all attacked devices while maintaining very low false positive rates. It maintains the same amount of attestations in the network, but enables switching between several attestation protocols according to the context; here, SEDA is not used during movement, but, when parked, the device can switch to SEDA to participate in the remote attestation aggregation. CRAFT-ASMP also enables heartbeats for any protocol, which also increases security by maintaining continuous attestation.

In terms of performance, the results show that CRAFT-ASMP provides a similar level of performance to CRAFT. This indicates that the ASMP features add security and flexibility to the framework at no performance cost. Comparing CRAFT-ASMP with the standalone protocols of SEDA+AID framework shows that CRAFT-ASMP reduces the data overhead generated by the framework, with a significant decrease for some scenarios. It also shows that, while antennas send most of the packets, the ratio of data sent by antennas is smaller with CRAFT-ASMP than with SEDA+AID. With regard to computational aspects, the number of computed HMACs is only slightly larger using CRAFT-ASMP than SEDA+AID. This is because of the heartbeat messages added in CRAFT (and, thus, CRAFT-ASMP), which improve security by maintaining continuous attestation, with the benefits outweighing the cost.

## 5. Conclusions

In this paper, we proposed ASMP (adaptive simultaneous multi-protocol) features to improve the CRAFT framework in terms of both security and flexibility. The ASMP features enable remote attestation frameworks, such as CRAFT, to simultaneously use multiple attestation protocols. Furthermore, these features enable devices to seamlessly switch from one protocol to another depending on the context, and, thus, select the most appropriate protocol in terms of both security and performance. Both stronger security and better performances were demonstrated using simulations in a real-world smart city environment and across multiple scenarios, in comparison to other existing remote attestation frameworks.

In future work, we plan to further improve CRAFT-ASMP flexibility with regard to protocol negotiation by implementing a specification language, such as Copland from Ramsdell et al. [23], as used by Helble et al. [22]. We also plan to propose a solution to enable continuous remote attestation for multiple owners of shared physical devices operating in different networks. Finally, using artificial intelligence, CRAFT-ASMP could more accurately adapt the protocol used, or even update the security parameters of the devices in real-time. 

## Figures and Tables

**Figure 1 sensors-23-04074-f001:**
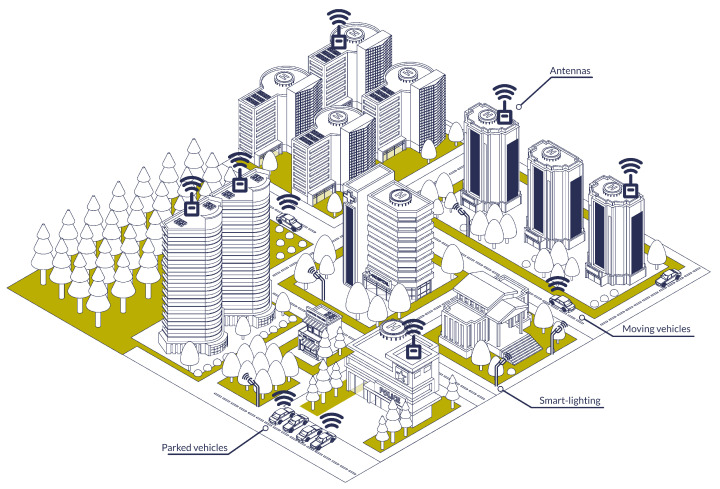
Smart city representation.

**Figure 2 sensors-23-04074-f002:**
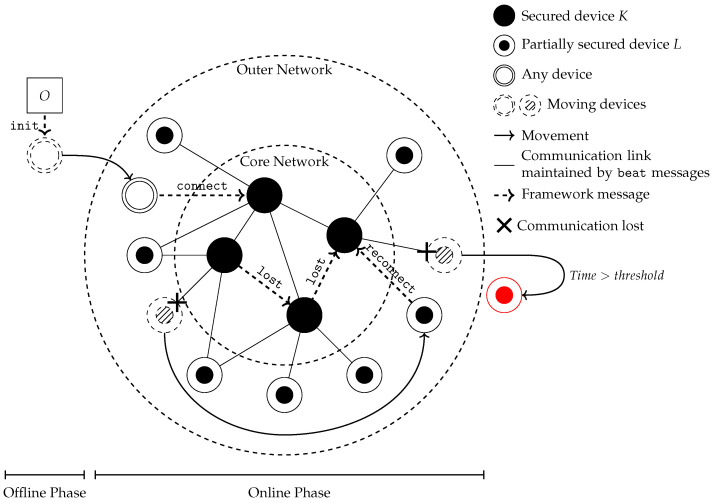
CRAFT overview.

**Figure 3 sensors-23-04074-f003:**
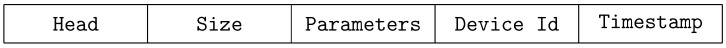
General CRAFT packet format.

**Figure 4 sensors-23-04074-f004:**
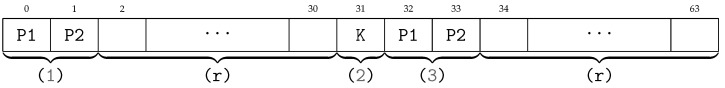
Detailed Parameters field.

**Figure 5 sensors-23-04074-f005:**
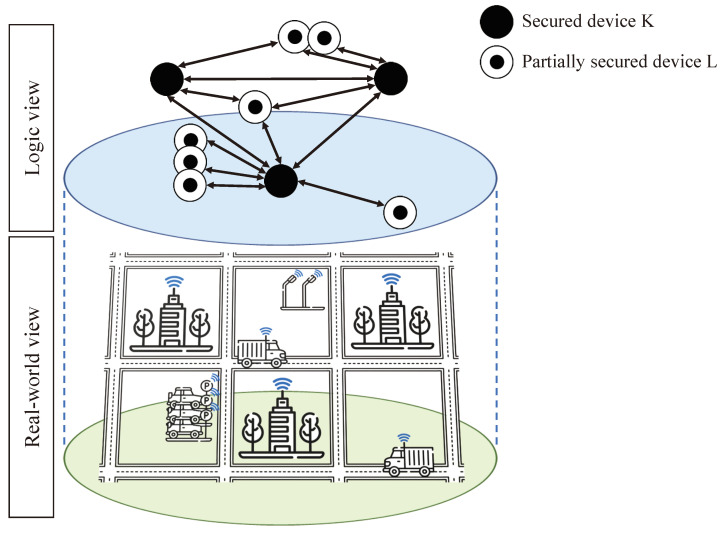
Simulation environment representation with both real-world and logic views.

**Figure 6 sensors-23-04074-f006:**
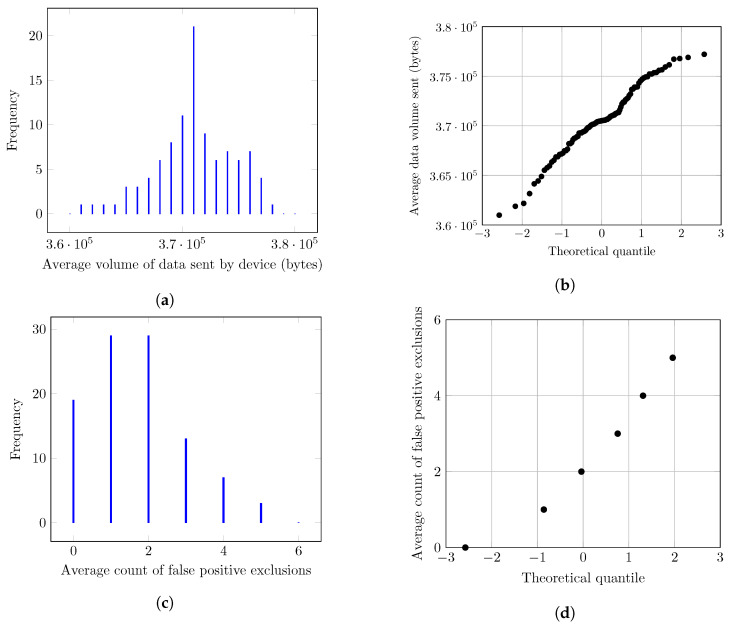
Graphs showing that simulation average results follow an approximately normal distribution. (**a**) Frequency of the average volume of data sent by devices in the *Basic* scenario using the CRAFT-ASMP framework over 100 iterations. (**b**) Q-Q plot of the average volume of data sent by devices in the *Basic* scenario using the CRAFT-ASMP framework over 100 iterations. (**c**) Frequency of false positive exclusions of mobile devices in the *Parking+Out* scenario using the CRAFT-ASMP framework over 100 iterations. (**d**) Q-Q plot of false positive exclusions of mobile devices in the *Parking+Out* scenario using the CRAFT-ASMP framework over 100 iterations.

**Figure 7 sensors-23-04074-f007:**
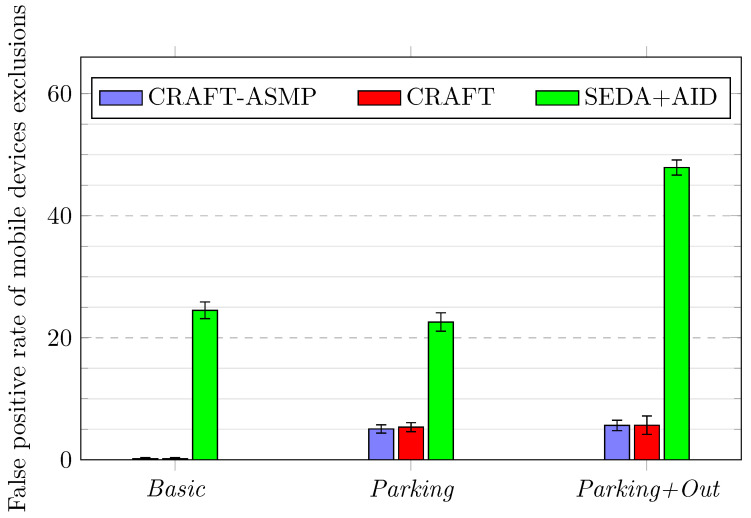
Frameworks comparison over three scenarios for false positive rates for mobile device exclusions.

**Figure 8 sensors-23-04074-f008:**
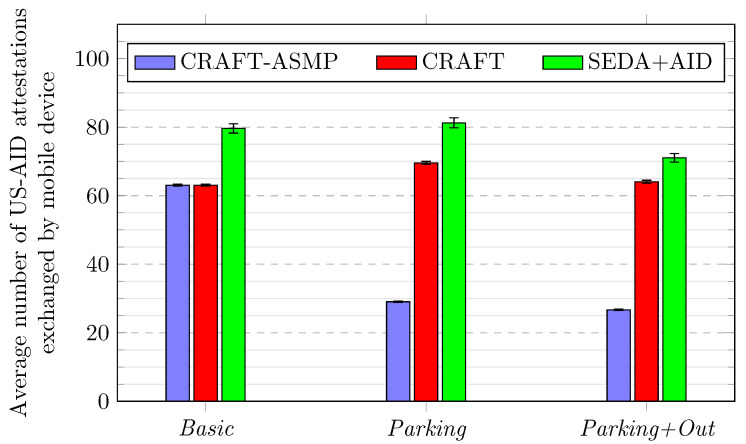
Frameworks comparison over three scenarios for US-AID attestation counts.

**Figure 9 sensors-23-04074-f009:**
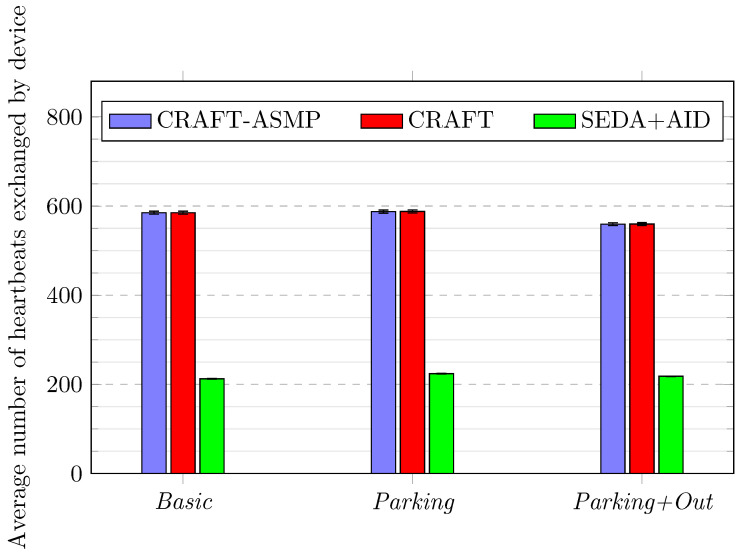
Frameworks comparison over three scenarios for heartbeat counts.

**Figure 10 sensors-23-04074-f010:**
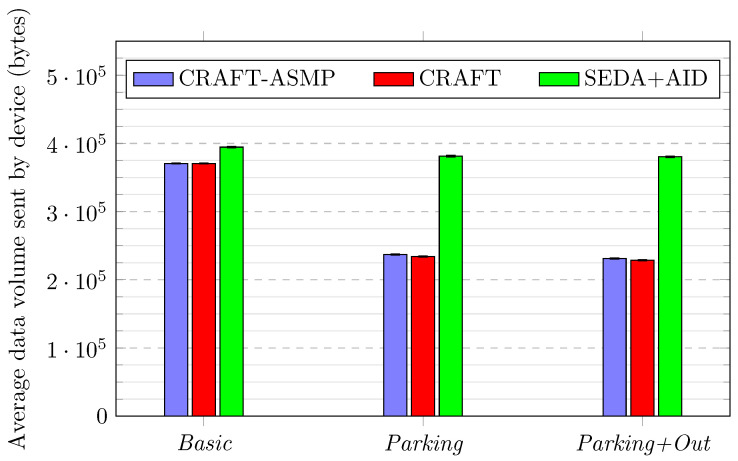
Frameworks comparison over three scenarios for data volume.

**Figure 11 sensors-23-04074-f011:**
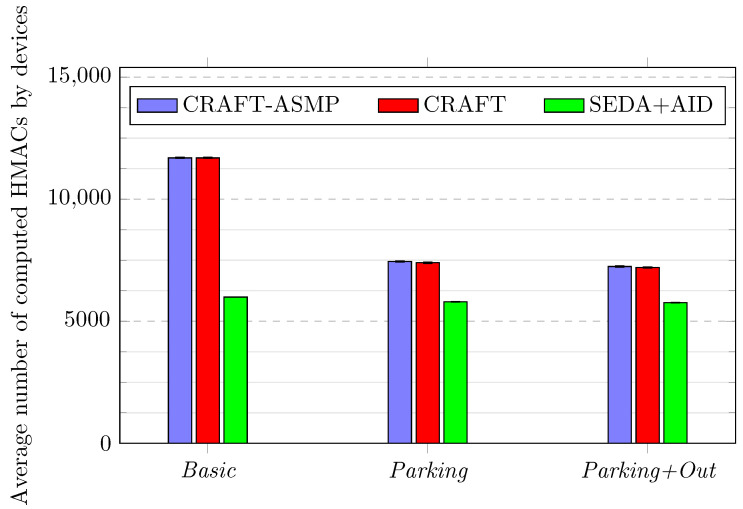
Frameworks comparison over three scenarios regarding the number of HMAC computations.

**Table 1 sensors-23-04074-t001:** Omnet++ simulation parameters.

Parameter	Value
Grid height and width (m)	600
Grid block height and width (m)	100
Device density (devices per km${}^{2}$)	278
Number of antennas	18
Number of static devices	42
Number of mobile devices	40
Antennas’ communication range (m)	300
Other devices’ communication range (m)	200
Antennas’ positioning	Chosen
Fixed devices’ positioning	Random
Mobile devices’ initial positioning	Random, at grid intersections
Mobile devices’ mobility model	Customized MovingMobilityBase
Simulation duration (s)	86,400
Radio model	UnitDiskRadio
Wireless interface model	WirelessInterface

**Table 2 sensors-23-04074-t002:** Omnet++ scenario parameters.

Parameter		Value	Scenario		
			Basic	Parking	Parking+Out
	Attestation frequency—$T_{a_{i}}$ (s)	3600	✓	✓	✓
	Heartbeat frequency—$T_{b_{i}}$ (s)	1100	✓	✓	✓
	Delay between two parking phases (s)	14,400		✓	✓
	Parking phase duration (s)	10,800		✓	✓
Excluded devicesthat should reconnect	Exclusion duration (s)	2000			✓
	Number of excluded static devices	10			✓
	Number of excluded mobile devices	10			✓
Excluded devices that should NOT reconnect	Exclusion duration (s)	7200			✓
	Number of excluded static devices	10			✓
	Number of excluded mobile devices	10			✓

**Table 3 sensors-23-04074-t003:** Frameworks comparison over three scenarios for static and mobile device exclusions.

Scenario	Framework	Mobile Devices Exclusions Rate		Static Devices Exclusions Rate	
		True Positive	False Positive	True Positive	False Positive
*Basic*	CRAFT-ASMP	-	0.20±0.15%	-	-
	CRAFT	-	0.20±0.15%	-	-
	SEDA+AID	-	24.50±1.37%	-	-
*Parking*	CRAFT-ASMP	-	5.05±0.68%	-	-
	CRAFT	-	5.35±0.73%	-	-
	SEDA+AID	-	22.58±1.51%	-	-
*Parking+Out*	CRAFT-ASMP	100±0%	5.63±0.85%	100±0%	0±0%
	CRAFT	100±0%	5.67±0.90%	100±0%	0±0%
	SEDA+AID	100±0%	47.90±1.23%	0±0%	0±0%

**Table 4 sensors-23-04074-t004:** Frameworks comparison over three scenarios for SEDA attestation counts.

Scenario	Framework	SEDA Attestation Count	
		Static Devices	Mobile Devices
*Basic*	CRAFT-ASMP	23.00±0.00%	0.00±0.00%
	CRAFT	23.00±0.00%	-
	SEDA+AID	23.00±0.00%	-
*Parking*	CRAFT-ASMP	23.00±0.00%	6.39±0.07%
	CRAFT	23.00±0.00%	-
	SEDA+AID	23.00±0.00%	-
*Parking+Out*	CRAFT-ASMP	21.33±0.00%	5.89±0.07%
	CRAFT	21.33±0.00%	-
	SEDA+AID	21.81±0.00%	-

**Table 5 sensors-23-04074-t005:** Ratio of data sent by different types of devices for each framework over three scenarios.

Scenario	Framework	Ratio of Data Sent by Type of Devices		
		Antenna	Static	Mobile
*Basic*	CRAFT-ASMP	92.46%	3.66%	3.88%
	CRAFT	92.46%	3.66%	3.88%
	SEDA+AID	96.31%	1.75%	1.94%
*Parking*	CRAFT-ASMP	87.81%	5.72%	6.47%
	CRAFT	88.40%	5.79%	5.81%
	SEDA+AID	96.04%	1.81%	2.15%
*Parking+Out*	CRAFT-ASMP	87.94%	5.46%	6.60%
	CRAFT	88.51%	5.52%	5.97%
	SEDA+AID	96.22%	1.72%	2.06%

## Data Availability

The data presented in this study are available on request from the corresponding author.

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
