# Peer review of "An Adaptive Simultaneous Multi-Protocol Extension of CRAFT"

_sensors, 2023, doi:10.3390/s23084074_

Round 1
Reviewer 1 Report
In this paper, the authors propose ASMP (Adaptative Simultaneous Multi-Protocol) features. These features fully enable the use of multiple remote attestation protocols for any device. They also enable devices to seamlessly switch protocols depending on factors such as environment, context, and neighboring devices. A comprehensive evaluation of these features in a real-world scenario and use cases demonstrates that they improve CRAFT’s flexibility and security with minimal impact on performance. Overall, the topic is timely, and the paper is written well, I have some comments which I believe will further improve the paper.
The abstract could be written in a more technical way. For example, a brief introduction of the topic that you are investigating, an explanation of why the topic is important in your field, what were the open gaps, your proposed system, method, and approach, and the key findings.
There is no single mathematical equation in the paper. If possible, add some essential equations to the revised paper.
Related work could be further improved.
The parameters used for simulation need to be justified.
The results need to be compared with any recent work in the literature.
Some typos and grammar errors are also identified.
Reviewer 2 Report
The following issues should be commented in order to improve the manuscript.
CRAFT-ASMP seems like a negligible improvement in certain aspects over CRAFT and significant improvement over SEDA+AID. Is the combination of SEDA+AID the best possible for competing with CRAFT?
It is unclear which (if any) type of encryption was used before HMAC, and how it would affect security, performance and power consumption. See for example, Table 4 [4].
Is a threshold static, static scenario-specific, dynamic or dynamic scenario-specific? In other words, is a threshold chosen manually in advance, or automatically during communication?
0% +/- 0% should be checked in Table 3, second column from the right, last line.
Is it possible to combine attestations with all communication messages, like it is sufficient that two devices understand each other? For example, if two devices do not understand each other due to wrong encryption keys, it is a clear attestation that one of them doesn't belong to the network. Having one device communicating with the others, and the another one not communicating with others seems like a clear case of attestation.
Reviewer 3 Report
Very well structured paper, presentation was good including the figures.
Well done. Topic is very interesting and the authors done well.
Round 2
Reviewer 1 Report
I have no further comments.